# The Atypical Manifestation of Pulmonary Tuberculosis in Patients with Bronchial Anthracofibrosis

**DOI:** 10.3390/jcm11195646

**Published:** 2022-09-25

**Authors:** Min Kyung Jung, Sang Young Lee, Jeong Min Ko

**Affiliations:** Department of Radiology, St. Vincent’s Hospital, College of Medicine, The Catholic University of Korea, Seoul 06591, Korea

**Keywords:** pulmonary tuberculosis, bronchial anthracofibrosis, computed tomography

## Abstract

It has been stated that bronchial anthracofibrosis (BAF) has an important relationship with pulmonary tuberculosis (TB), and the coexistence of TB and BAF is high. The purpose of this study was to compare the differences in computed tomography (CT) characteristics of pulmonary TB according to the presence of underlying BAF. Total of 202 consecutive patients who were diagnosed with pulmonary TB and underwent bronchoscopy and CT in our institution were retrospectively reviewed. We classified the patients into two groups according to the presence of BAF and compared the clinicoradiological findings between the two groups (anthracofibrosis group vs. nonanthracofibrosis group). Elderly and female patients were significantly higher in anthracofibrosis group (mean age 79 ± 7 (64–94) vs. 56 ± 17 (16–95), *p* < 0.001; female 89% vs. 29%, *p* < 0.001). The frequency of internal low-density area or focal contour bulge within atelectasis (64% vs. 1%, *p* < 0.001), lower lobe predominance (43% vs. 9%, *p* < 0.001), endobronchial involvement (46% vs. 15%, *p* < 0.001), and lymphadenopathy (57% vs. 28%, *p* = 0.002) were significantly higher in anthracofibrosis group. In contrast, the anthracofibrosis group showed lower frequency of upper lobe predominance (32% vs. 81%, *p* < 0.001) and cavitation (14% vs. 51%, *p* = 0.001). In conclusion, being aware of these atypical manifestations of pulmonary TB in the presence of BAF will be of great help in early detection of TB.

## 1. Introduction

Bronchial anthracofibrosis (BAF) is a bronchoscopic diagnosis defined as bronchostenosis, proximal airway narrowing or obliteration, associated with bluish-black muscosal anthracotic pigmentation with or without a history of occupational dust exposure; however, it usually occurs in persons with a long-standing history of biomass fuel smoke exposure [1,2,3,4,5,6,7,8,9,10,11]. BAF has an important relationship with pulmonary tuberculosis (TB). For this reason, the coexistence of TB and BAF is reported to be high (20–61%) [3,12].

Micronodules, consolidation/macronodule, cavitation, and tree-in-bud opacities are well-known typical computed tomography (CT) findings of active pulmonary TB [13,14,15,16,17]. However, there are cases with unusual or atypical CT findings which makes its diagnosis difficult. It is known that patients with diabetes mellitus, idiopathic pulmonary fibrosis, or systemic lupus erythematosus show unusual or atypical radiologic manifestations of pulmonary TB [18,19,20,21]. We experienced that pulmonary TB with BAF also showed distinctive radiologic manifestations. To evaluate the effect of BAF on CT characteristics of pulmonary TB, we performed this retrospective study. To our knowledge, this study is the first comparison of the CT findings of active pulmonary TB in anthracofibrosis and nonanthracofibrosis groups.

## 2. Materials and Methods

### 2.1. Patients

A total of 308 consecutive patients who were diagnosed with active pulmonary TB and underwent CT scans in our institution from January 2017 to November 2019 were retrospectively reviewed. Out of those patients, 71 cases who did not undergo bronchoscopy were excluded. Among the 237 patients, 19 were excluded due to co-existing pulmonary disease (pneumonia, *n* = 6; nontuberculous mycobacterial infection, *n* = 2; pulmonary edema, *n* = 1; severe emphysema, *n* = 2; interstitial lung disease, *n* = 1; pneumoconiosis, *n* = 2; lung collapse due to extensive pleural TB, *n* = 1; lung cancer, *n* = 2, lung metastasis from malignancy except for lung cancer, *n* = 2). We also excluded 2 patients with extensive old TB lesions on CT scans because extensive old lesions made it impossible to assess the CT findings of active pulmonary TB due to severe lung destruction. Among the 216 patients, 14 cases who were bacillary negative, but showed radiological and clinical improvement after administration of antituberculous drugs were excluded. Finally, 202 patients with active pulmonary TB were enrolled for evaluation (126 men and 76 women; mean age, 59 years; age range, 16–95 years).

We analyzed the medical records, microbiology study results and pulmonary parenchymal changes on CT scans. The diagnosis of active pulmonary tuberculosis, with or without tuberculous pleurisy, was based on positive results for acid-fast bacilli (AFB) staining or growth of *Mycobacterium tuberculosis* from sputum and bronchial washes; positive results on polymerase chain reaction (PCR) test for *M. tuberculosis*; or histologic confirmation.

### 2.2. CT Protocol

Conventional CT with or without intravenous administration of contrast medium (90 mL for 65 kg at 2–2.5 mL/s) and high-resolution CT (HRCT) was obtained with three MDCT scanners. The decision to perform contrast enhancement was made by the attending clinicians based on their suspicion of malignancy or TB lymphadenitis from chest radiographs of the patients. The parameters for the SOMATOM Definition Flash scanner (Siemens Healthcare, Forchheim, Germany) were: detector collimation, 128 × 0.6 mm; rotation time, 500 ms; pitch, 1.3 for conventional CT and 1.5 for HRCT; 100-kV tube voltage; automatic tube current modulation for conventional CT. Parameters for the Discovery CT750 HD scanner and Optima CT660 (General Electric Medical Systems, Milwaukee, WI, USA) were: detector collimation, 64 × 0.625 mm; rotation time, 600 ms for conventional CT and 500 ms for HRCT; pitch, 1.375 for conventional CT and 0.984 for HRCT; 100-kV tube voltage; automatic tube current modulation for both conventional CT and HRCT. For conventional CT, we analyzed axial images with 1 mm slice thickness at 1 mm intervals and a high-spatial-frequency algorithm. For HRCT, we reviewed the axial images with 1 mm slice thickness at 3 mm intervals with a high-spatial-frequency algorithm.

### 2.3. Image Analysis

CT scans obtained before administration of anti-tuberculous medication were analyzed for the presence and distribution of micronodules. Micronodules were defined as small rounded opacities with a diameter of less than 7 mm. The distribution of micronodules was classified as centrilobular (micronodules limited to centrilobular regions), perilymphatic (micronodules along the bronchovascular and centrilobular interstitium, interlobular septa, and subpleural region), or random (uniform distribution of micronodules throughout the lung) [22,23,24]. We also assessed the presence or absence of tree-in-bud, large opacity (consolidation or macronodule), cavitation, ground glass opacity, bronchovascular bundle thickening, interlobular septal thickening, atoll sign, galaxy/cluster sign, lymphadenopathy (short-axis diameter > 1 cm or central necrosis) and endobronchial involvement (bronchial wall thickening or peribronchial cuff of soft tissue, bronchial involvement with tuberculous lymphadenitis, intraluminal low-density polypoid mass with narrowing), which are well-known common CT features of active pulmonary and endobronchial TB [17,25,26,27]. In addition, the presence of internal low-density area (on CT with contrast enhancement) or focal bulging contour (on CT with/without contrast enhancement) within atelectasis and zonal predilection were assessed. It was defined as upper lobe predominance when the upper lobe and superior segment of the lower lobe were mainly involved; as lower lobe predominance if the middle lobe and basal segment of the lower lobe were mainly involved. The lingula was defined as the middle lobe. It was defined as no zonal predominance when neither upper lobe predominance nor lower lobe predominance was applicable.

All CT scans and medical records were retrospectively reviewed by three thoracic radiologists (with 11, 18, and 2 years of experience in thoracic imaging, respectively). The three radiologists independently and blindly analyzed the CT scans for the presence or absence of TB features in consensus.

### 2.4. Statistical Analysis

We divided the patients into two groups according to the presence of BAF (anthracofibrosis group vs. nonanthracofibrosis group) and compared the clinical and CT findings between these two groups. BAF, defined as bronchostenosis, proximal airway narrowing or obliteration, associated with bluish-black muscosal anthracotic pigmentation with or without a history of occupational dust exposure, was diagnosed via bronchoscopy [1,2,3,4,5,6,7,8,9,10,11]. Comparisons between the two groups were performed using independent t test for continuous variables and chi-square or Fisher exact test for categoric variables. A *p* value of <0.05 was considered statistically significant. 

### 2.5. Ethics Statement

This study was conducted in accordance with the Declaration of Helsinki, and approved by The Catholic university of Korea St. Vincent’s Hospital Institutional Review Board (protocol code: VC21RESI0259, and date of approval: 14 December 2021). The informed consent was waived due to its retrospective design. 

## 3. Results

### 3.1. Patients

Clinical features, microbiology results and frequency of CT findings are summarized in Table 1 and Table 2. Of the 202 patients, 97 patients suffered from chronic illness including renal failure (*n* = 4), malignancy (*n* = 17), acquired immune deficiency syndrome (*n* = 1), diabetes (*n* = 58), collagen vascular disease (*n* = 5), chronic obstructive pulmonary disease or asthma (*n* = 5), and liver disease (*n* = 17). Diagnosis of active pulmonary TB was confirmed by positive sputum microbiology results (acid-fast bacilli (AFB), *n* = 49; polymerase chain reaction (PCR), *n* = 60; culture, *n* = 73) in 87 patients. One hundred eighty-two patients were diagnosed by positive microbiology results from bronchial lavage (AFB, *n* = 68; PCR, *n* = 153; culture, *n* = 177). Of the 202 patients, 195 were microbiologically positive (AFB or PCR or culture) from airway secretions (sputum or bronchial washing fluid). Seven were microbiologically negative from airway secretions but were pathologically diagnosed.

The overall frequency of micronodules, consolidation/macronodule, cavitation, and tree-in-bud opacities were 93%, 91%, 46%, and 43%, respectively. Bronchovascular bundle thickening and interlobular septal thickening were also commonly seen (78% and 68%, respectively). Lymphadenopathy and endobronchial involvement were relatively infrequent (32% and 19%, respectively). The frequency of internal low-density or focal bulging area within atelectasis was 10%. Upper lobe predominance (74%) was more prevalent compared to lower lobe predominance (13%).

### 3.2. Clinical Features and Microbiology Results of Pulmonary TB According to the Presence of BAF

Of the 202 patients with active pulmonary TB, 28 and 174 were classified into anthracofibrosis and nonanthracofibrosis groups, respectively. Elderly and female patients were significantly higher in the anthracofibrosis group compared to the nonanthracofibrosis group (mean age 79 ± 7 (64–94) vs. 56 ± 17 (16–95), *p* < 0.001; female 89% vs. 29%, *p* < 0.001). Current or past history of cigarette smoking was significantly lower in anthracofibrosis group (7% vs. 53%, *p* < 0.001). The frequency of co-morbidity and microbiology results did not show significant differences between the two groups, except for bronchial lavage PCR positivity for *M. tuberculosis* (96% vs. 74%, *p* = 0.008).

### 3.3. CT Findings of Pulmonary TB According to the Presence of BAF

The proportion of conventional CT with enhancement, conventional CT without enhancement, and HRCT were 89%, 10%, and 1%, respectively. Anthracofibrosis group showed 96%, 4%, and none, and nonanthracofibrosis group showed 87%, 11%, 2%, respectively.

Common CT findings of anthracofibrosis group were micronodules (89%), consolidation/macronodule (89%), bronchovascular bundle thickening (68%), and internal low-density area or focal contour bulge within atelectasis (64%) in order. The frequency of lymphadenopathy (57% vs. 28%, *p* = 0.002), endobronchial involvement (46% vs. 15%, *p* < 0.001), and lower lobe predominance (43% vs. 9%, *p* < 0.001) were significantly higher in the anthracofibrosis group. In particular, atelectasis with internal low-density or focal bulging area (64% vs. 1%, *p* < 0.001) was significantly higher in the anthracofibrosis group, whereas it showed significantly lower frequency of cavitation (14% vs. 51%, *p* = 0.001) and upper lobe predominance (32% vs. 81%, *p* < 0.001). Cases with distinctive CT findings of pulmonary TB in the presence of BAF are seen in Figure 1, Figure 2 and Figure 3.

## 4. Discussion

BAF is related to biomass combustion products inhalation with or without a history of occupational dust exposure. BAF occurs commonly among the Asian and Black populations. It is especially common among non-smoking older women who have spent long hours cooking on firewood or soft coals while being exposed to dense smoke in poorly ventilated kitchens. Ciliary movement removes most inhaled anthracotic particles, but residual particles can accumulate at the branching points of the airway. Anthracotic pigmentation itself does not induce focal bronchial narrowing because carbon is inert. However, infection, air pollutants or smoke can induce healing and fibrosis of bronchial mucosa with anthracotic pigmentation resulting in BAF [1,2,3,4,5,6,28,29,30,31,32].

The first publication in the English literature regarding BAF was made in 1998 on the journal CHEST [1]. Thereafter, several descriptions on radiological findings have been published. At the time of the initial publication by Dr. Chung et al. [1], the concurrent presence of two diseases (BAF and pulmonary TB) was seen in 17 TB (61%) of 28 BAF patients. It is not known exactly what the interrelationship is between the two diseases. At the time of Dr. Chung’s first description, endobronchial TB may be the cause of anthracofibrosis of the airways, but black pigmentation and airway narrowing is not always seen in endobronchial TB. These findings are more frequently seen in pneumoconiosis or in individuals exposed to biomass smoke. Park et al. [3] found that peribronchial and mediastinal lymphadenopathy, involvement of more lung lobes, bilateral lung involvement and stenosis of any lobe of the right lung were significantly more common in BAF patients than in endobronchial TB patients. It was also found that patients with endobronchial TB displayed contiguous luminal narrowing in the main and lobar bronchi, whereas the main bronchus tended to be unaffected in patients with BAF. These differences in CT findings between BAF and endobronchial TB suggest that TB may not be a causative factor in BAF. Nowadays, TB is considered as an associated condition rather than a causative factor [2,3].

In our study, BAF was coexistent in 14% of all TB patients. Non-smoking, elderly and female patients were significantly higher in the anthracofibrosis group compared to nonanthracofibrosis group (cigarette smoking 7% vs. 53%, *p* < 0.001; mean age 79 ± 7 (64–94) vs. 56 ± 17 (16–95), *p* < 0.001; female 89% vs. 29%, *p* < 0.001). These results are related to the fact that BAF is more common among non-smoking older women, which is associated with the pathogenesis of BAF [1,2,3,4,5,6,28,29,30,31,32]. To our knowledge, no studies have been published showing that the CT findings of pulmonary TB in the elderly, females, and non-smokers are different. Therefore, it is unlikely that sex, age, and smoking history acted as bias.

In our study, the CT findings of pulmonary TB in the presence of BAF did not show typical CT findings. The frequency of cavitation (14% vs. 51%, *p* = 0.001) was significantly lower in anthracofibrosis group. Similarly, Hwang et al. [6] reported a low frequency of cavitation (1/10) in bronchoscoped pulmonary TB patients with BAF. Cavitation is one of the important CT findings of active pulmonary TB [13,14,15]. We believe that the reason why cavitation is less frequent in anthracofibrosis group compared to the nonanthracofibrosis group is that caseous necrosis of pulmonary TB could not escape through bronchus due to bronchostenosis of BAF. In BAF, atelectasis may occur as a secondary finding due to bronchostenosis [2,3,6,33,34]. Therefore, it is thought that caseous necrosis appeared as internal low-density or focal bulging area within atelectasis instead of cavitation. Thus, internal low-density or focal bulging area within atelectasis (64% vs. 1%, *p* < 0.001) is an important key CT feature suggesting pulmonary TB in the presence of BAF; this could be a pitfall because it is an atypical imaging feature of pulmonary TB.

Upper lobe predominance is also a typical CT finding of TB [15,16,17,18]. However, lower lobe predominance (43%) was more common than upper lobe predominance (32%) with underlying BAF in our study; this could be another pitfall. It has been stated that the coexistence of TB and BAF is high (20–61%) because of an important relationship [3,12]. Therefore, it is reasonable to assume that TB is likely to develop in lobes affected by BAF. BAF occurs in multifocal points, especially in the right middle lobe [2,3,33,34]. The reason why BAF tends to affect the middle lobe predominantly is due to ineffective collateral ventilation and impaction of carbon particles at the site of bronchial bifurcation around the right middle lobe orifice due to sharp angular take-off of the middle lobe bronchus. Subsequently, inflammatory response results in fibrosis and stenosis of the affected bronchus [2,9,35,36]. Because of this zonal predilection of BAF, TB in the presence of BAF is thought to have shown lower lobe predominance compared to upper lobe predominance. 

The CT finding of endobronchial involvement (46% vs. 15%, *p* < 0.001) was significantly higher in anthracofibrosis group in our study. The exact pathogenesis of endobronchial TB is not yet understood; however, suggested possible mechanisms of infection include (a) direct extension from an adjacent parenchymal focus, (b) implantation of organisms from the infected sputum, (c) hematogenous dissemination, (d) lymph node erosion into a bronchus and (e) spread of infection via the lymphatics [37,38]. It is known that exposure to toxic substances in biomass smoke or the pathology of BAF itself leads to the compromise of pulmonary and bronchial immune defense mechanisms. Therefore, *M. tuberculosis* implantation can easily occur in any origin if BAF is present (a–e, as described above) [7,39,40]. The high frequency of lymphadenopathy (57% vs. 28%, *p* = 0.002) in the anthracofibrosis group may also be due to the compromised local immune defense mechanism. However, hyperdense or calcified mediastinal, hilar and peribronchial lymph nodes and lymph node enlargement are the main findings in BAF [2,33,34]. Therefore, BAF-associated lymphadenopathy and TB lymphadenitis may coexist, leading to possible overestimation of the results. In our study, lymphadenopathy was defined as lymph node with a short-axis diameter of 1 cm or more, or lymph node with central necrosis on CT images. Histological examination of the lymph node was not performed in our study. There were 8 patients (29%) without both lymphadenopathy and endobronchial involvement, and 9 patients (32%) with both lymphadenopathy and endobronchial involvement. 

The reason why bronchial lavage PCR positive for *M. tuberculosis* was statistically higher in anthracofibrosis group (96% vs. 74%, *p* = 0.008) is thought to be due to the high frequency of endobronchial TB in the anthracofibrosis group. Furthermore, there is a difference in TB positivity using either sputum or bronchial washes as samples in anthracofibrosis group. It may be because patients of BAF are more likely to expectorate sputum incorrectly due to their bronchostenosis. Therefore, the collection of appropriate samples via bronchial washes rather than sputum can be more helpful for disease diagnosis in patients with BAF.

The limitations of our study must be acknowledged. First, the comparison group is not all other pulmonary TB patients without BAF but all pulmonary TB patients without BAF who had undergone bronchoscopy. Since BAF can be diagnosed only by bronchoscopy, pulmonary TB patients who underwent bronchoscopy and did not have BAF had to be set as the comparison group. If all other pulmonary TB patients were used as the comparison group, pulmonary TB patients with BAF could also have been included. The bronchoscoped pulmonary TB patients without BAF may or may not be representative of all non-BAF pulmonary TB patients. However, our comparison group showed radiologic findings similar to other studies. In other studies [16,17,18], the most common CT findings of pulmonary TB were consolidation and micronodules in upper lobes and superior segments of lower lobes (upper lobe predominance), and the frequency of cavitation was 20–66%. In our study, the most common CT findings in nonanthracofibrosis group were micronodules (93%) and consolidation/macronodule (89%). The frequency of cavitation and upper lobe predominance was 48%, and 80%, respectively. It is not clear what prompted the need for bronchoscopy in non-BAF patients. The bronchoscoped TB patients may have expectorated sputum incorrectly, or the sputum test may have been negative, or the attending clinician may have decided to perform bronchoscopy initially. Second, we retrospectively reviewed the medical records and could not report the environmental or occupational exposure history although it is related to BAF and biomass fuel smoke exposure.

## 5. Conclusions

Active pulmonary TB with BAF showed a high prevalence of internal low-density foci or focal contour bulge within atelectasis, lower lobe predominance, endobronchial involvement, and lymphadenopathy, whereas it showed a low frequency of cavitation. The CT findings of pulmonary TB with BAF are quite distinctive from the typical features of pulmonary TB without BAF. Therefore, being aware of these distinctive CT findings of pulmonary TB in the presence of BAF will be of great help in the early detection of TB.

## Figures and Tables

**Figure 1 jcm-11-05646-f001:**
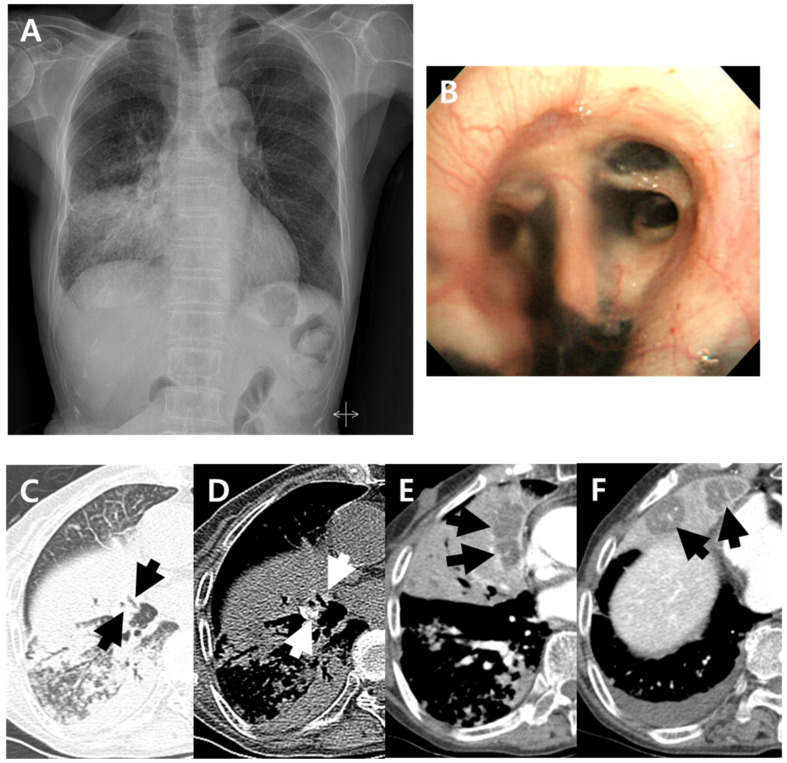
A 79-year-old woman with pulmonary tuberculosis and underlying bronchial anthracofibrosis presented with an abnormal chest plain radiograph. (**A**) Chest plain radiograph showed consolidation and ill-defined nodular opacities in right lower lung field with lower lobe predominance. (**B**) Bronchoscopic image showed luminal narrowing at segmental bronchi of right middle lobe with multifocal deposition of black pigmentation, and bronchial anthracofibrosis was diagnosed. (**C**,**D**) Non-enhanced axial CT scan showed bronchostenosis (black arrows) at medial and lateral segmental bronchi of right middle lobe and peribronchial hyperattenuation (white arrows) around medial and lateral segmental bronchi of right middle lobe. These CT findings suggested bronchial anthracofibrosis. (**E**,**F**) Contrast-enhanced axial CT scan showed internal low-density areas (arrows) within atelectasis of right middle lobe. These findings were atypical CT findings of pulmonary tuberculosis, which cannot be suspected of TB. By the way, clustered micronodules with tree-in-buds and consolidations in right lower lobe were shown on the axial CT scan which were well known typical CT findings of pulmonary tuberculosis. CT = computed tomography.

**Figure 2 jcm-11-05646-f002:**
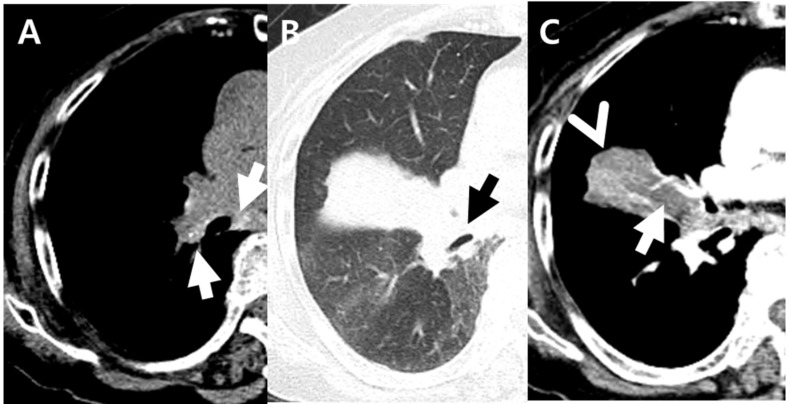
An 87-year-old woman with pulmonary tuberculosis and underlying bronchial anthracofibrosis presented with dyspnea. (**A**,**B**) Non-enhanced axial CT scan showed tiny hyperdense foci (white arrows) within enlarged peribronchial and mediastinal nodes and complete obstruction (black arrow) of right middle lobar bronchus, resulting in atelectasis of right middle lobe. These CT findings suggested bronchial anthracofibrosis. (**C**) Contrast-enhanced axial CT scan showed focal contour bulge (arrowhead) and internal low-density area (arrow) within atelectasis. These findings were the only CT findings suggesting pulmonary tuberculosis in this patient. Culture of *Mycobacterium tuberculosis* and polymerase chain reaction test for *Mycobacterium tuberculosis* were positive from bronchial washes. CT = computued tomography.

**Figure 3 jcm-11-05646-f003:**
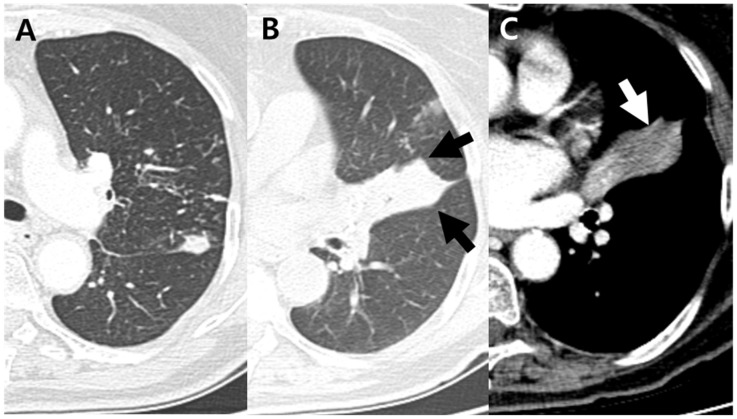
A 75-year-old woman with pulmonary tuberculosis and underlying bronchial anthracofibrosis presented with cough and sputum. (**A**) Axial CT scan showed clustered micronodules and consolidations in left upper lobe. These findings were well known typical CT findings of pulmonary tuberculosis. (**B**,**C**) Axial CT scan showed complete obstruction of lingular divisional bronchus, resulting in atelectasis with focal contour bulge (black arrows) and internal low-density area (white arrow). These findings were atypical CT findings of pulmonary tuberculosis, which cannot be suspected of TB. CT = computed tomography.

**Table 1 jcm-11-05646-t001:** The clinical features, microbiology results, and statistical differences between anthracofibrosis group and nonanthracofibrosis group.

	Total	Anthracofibrosis Group	Nonanthracofibrosis Group	*p* Value ^a^
Number of patients	202	28	174	
Age (mean ± SD, range)	59 ± 18 (16–95)	79 ± 7 (64–94)	56 ± 17 (16–95)	<0.001 ***^b^
Sex (M:F)	126:76	3:25	123:51	<0.001 ***
Smoking history, M:F	95 (47%), 90:5	2 (7%), 2:0	93 (53%), 88:5	<0.001 ***
Co-morbidity	97 (48%)	11 (39%)	86 (49%)	0.319
Chronic renal failure	4 (2%)	1 (4%)	3 (2%)	0.452
Malignancy	17(8%)	4 (14%)	13(7%)	0.264
AIDS	1 (0.5%)	0	1 (0.6%)	1.000
DM	58(29%)	6 (21%)	52(30%)	0.359
Collagen vascular disease	5 (2%)	0	5 (3%)	1.000
COPD, asthma	5 (2%)	1 (4%)	4 (2%)	0.530
Liver disease	17 (8%)	0	17 (10%)	0.137
Sputum				
AFB positivity	49/130 (38%)	6/18 (33%)	43/112 (38%)	0.681
AFB number, mean ± SD	0.86 ± 1.26	0.56 ± 0.86	0.91 ± 1.31	0.143 ^b^
PCR positivity	60/111 (54%)	8/17 (47%)	52/94 (55%)	0.529
Culture positivity	73/115 (63%)	11/15 (73%)	62/100 (62%)	0.395
Bronchial washing fluid				
AFB positivity	68/198 (34%)	12/27 (44%)	56/171 (33%)	0.234
AFB number, mean ± SD	0.73 ± 1.12	0.78 ± 1.01	0.73 ± 1.14	0.822 ^b^
PCR positivity	153/199 (77%)	27/28 (96%)	126/171 (74%)	0.008 **
Culture positivity	177/200 (88%)	27/28 (96%)	150/172 (87%)	0.211

Note: AIDS = acquired immune deficiency syndrome, DM = diabetes mellitus, COPD = chronic obstructive pulmonary disease, AFB = acid-fast bacilli, PCR = polymerase chain reaction. ^a^ Unless otherwise noted, the chi-square or Fisher exact test was used. ^b^ Independent *t* test was used. ** *p* < 0.01, *** *p* < 0.001.

**Table 2 jcm-11-05646-t002:** The frequency of CT findings and statistical differences between anthracofibrosis group and nonanthracofibrosis group.

	Total (*n* = 202)	Anthracofibrosis Group (*n* = 28)	Nonanthracofibrosis Group (*n* = 174)	*p* Value ^c^
Micronodule	187 (93%)	25 (89%)	162 (93%)	0.443
Centrilobular	106 (52%)	18 (64%)	88 (51%)	0.178
Perilymphatic ^a^	169 (84%)	21 (75%)	148 (85%)	0.179
*Peribronchovascular*	166 (82%)	21 (75%)	145 (83%)	0.292
*Septal*	103 (51%)	10 (36%)	93 (53%)	0.081
*Subpleural*	56 (28%)	9 (32%)	47 (27%)	0.573
Random	11 (5%)	2 (7%)	9 (5%)	0.652
Tree-in-bud	86 (43%)	15 (54%)	71 (41%)	0.205
Consolidation/Macronodule	183 (91%)	25 (89%)	158 (91%)	0.732
Cavitation	93 (46%)	4 (14%)	89 (51%)	0.001 **
Ground glass opacity	39 (19%)	8 (29%)	31 (18%)	0.181
Bronchovascular bundle thickening	157(78%)	19 (68%)	138(79%)	0.176
Interlobular septal thickening	138 (68%)	16 (57%)	122 (70%)	0.171
Atoll sign	4 (2%)	0	4 (2%)	1.000
Galaxy/Cluster sign	14 (7%)	0	14 (8%)	0.225
Lymphadenopathy	65 (32%)	16 (57%)	49 (28%)	0.002 **
Endobronchial involvement ^b^	39 (19%)	13 (46%)	26 (15%)	<0.001 ***
Internal low-density area or focal contour bulge within atelectasis	20 (10%)	18 (64%)	2 (1%)	<0.001 ***
Upper lobe predominance	150 (74%)	9 (32%)	141 (81%)	<0.001 ***
Lower lobe predominance	27 (13%)	12 (43%)	15 (9%)	<0.001 ***
No zonal predominance	25 (12%)	7 (25%)	18 (10%)	0.056

^a^ Nodules were considered as perilymphatic when they were in at least one of peribronchovascular interstitium, interlobular septa, and subpleural region. ^b^ Endobronchial involvement was defined as bronchial wall thickening or peribronchial cuff of soft tissue, bronchial involvement with tuberculous lymphadenitis, or intraluminal low-density polypoid mass with narrowing. ^c^ The chi-square or Fisher exact test was used. ** *p* < 0.01, *** *p* < 0.001.

## Data Availability

Data available on request due to restrictions. The data presented in this study are available on request from the corresponding author.

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
