# Peer review of "The Atypical Manifestation of Pulmonary Tuberculosis in Patients with Bronchial Anthracofibrosis"

_jcm, 2022, doi:10.3390/jcm11195646_

Round 1

Reviewer 1 Report

The authors have undertaken a hospital-based (St. Vincent's Hospital, College of Medicine, the Catholic University of Korea) retrospective cohort study of pulmonary TB patients undergoing both CT scan and bronchoscopy, for the purpose of comparing the clinical, mycobacteriological and radiologic features in patients with and with bronchial anthracofibrosis (BAF). A total of 28 BAF-associated pulmonary TB patients were identified from within a total of 216 pulmonary TB patients. The radiologic description of these patients in particular is quite useful to the diagnosis of pulmonary TB with atypical features. Several comments/questions suggestions follow, in relative order of their importance.

1. The authors need to make clear that BAF is a bronchoscopic diagnosis - proximal airway narrowing or obliteration with blue/black airway hyperpigmentation, usually in persons exposed to occupational dust or biomass combustion products. It is not limited to patients exposed to biomass combustion products. It is also important to note that the comparison group in the study is not all other (non-BAF) adult pulmonary TB patients but all adult pulmonary TB patients that that had undergone bronchoscopy; most of these patients had upper lung zone disease; 50% were cavitary. The atypical presentation of the BAF-associated patients might have prompted their bronchoscopy; it is not clear what prompted the bronchoscopy in the non-BAF patients - was it to obtain an airway secretion? They may or may not be representative of all non-BAF adult pulmonary TB patients.

2. Ideally the authors need to state the disease type of their patients (new active vs relapse/retreatment). They excluded 2 with a past history of TB, but that is a very small relapse/retreatment rate. All of those with a past history of pulmonary TB should ideally be excluded as their past disease could complicate the interpretation of the imaging in the repeat (current) episode. It would also be ideal if the authors could report the smoking (cigarette) history of their patients. 

3. It is not clear from the authors results just how many of their patients were culture positive (or at minimum smear-positive with a positive PCR -  99% or more of smear-positive patients should grow the pathogen) from airway secretions (i.e. sputum or bronchoscopy wash or lavage).  For purposes of scientific rigour - i.e. it cannot be disputed that their patients had pulmonary TB - they really aught to consider excluding patients that did not meet these criteria, especially those that were bacillary negative or only culture positive from pleural fluid. 

4. The authors overlooked an important reference (Hwang J, et al. Bronchial anthracofibrosis and tuberculosis in immigrants to Canada from the Indian subcontinent. Int J Tuberc Lung Dis 2010; 14(2):231-37) which also reports on clinical, mycobacteriologic, radiologic and pathologic findings in BAF associated pulmonary TB. Of their BAF-associated pulmonary TB patients, 16(57%) had lymphadenopathy and 13(46%) had endobronchial involvement. Did any patients have neither?

5. In their image analysis section the authors refer to "CT scans obtained before administration of anti-TB drugs" being analyzed. Does this imply that some scans were performed after the start of anti-TB drugs? It would be helpful to know, in each group (BAF and non-BAF) when the CT scans were performed relative to the start date of treatment. If there was a big difference in the timing of the scans relative to the start date of treatment, it could affect their findings.

6. Intrathoracic adenopathy is a common feature in BAF. Enlarged nodes are better seen on an enhanced study. Some of their scans were enhanced, others were not. For purposes of comparing the frequency of lymphadenopathy in BAF-associated vs non BAF-associated pulmonary TB they should indicate the proportion of patients in each group whose CT scans were enhanced. Also, under the CT protocol section of the methods, do the authors mean to say "For conventional CT, we analyzed axial images with 1-mm slice thickness at 1-cm interval......"? 

7. The discussion should include a limitations section.

Reviewer 2 Report

The article entitled “The atypical manifestation of pulmonary tuberculosis in patients with bronchial anthracofibrosis” by Jung et al has discussed about the coexistence of TB and BAF. In the paper authors have discussed about differences in computed tomography (CT) characteristics of pulmonary TB according to the presence of underlying BAF. The article is well written and discussed about the points to be focussed while doing CT.

However, the authors need to elaborate more on below mentioned points before finally accepting the paper for publication:  

The paper has found high proportion of anthracosis and active TB among female and have discussed its corelation with biomass fuel smoke exposure.  However, no data has been shown in the submitted paper about the percentage of man and women who were exposed to smoke., or had history of cigarette smoking.

2. There is difference in TB positivity using either sputum and BAL as samples. Is it due to Anthracofibrosis? Authors need to discuss more on it as it may help in deciding the collection of appropriate sample for disease diagnosis in patients of Anthracofibrosis.

Round 2

Reviewer 1 Report

No comments to authors